# Cost of Climate Change: Risk of Building Loss from Typhoon in South Korea

**Ji-Myong Kim** [1] [image: ORCID], **Seunghyun Son** [2], **Sungho Lee** [3] **and Kiyoung Son** [4],*

[1] Department of Architectural Engineering, Mokpo National University, Mokpo 58554, Korea; jimy6180@gmail.com
[2] Department of Architectural Engineering, Kyung Hee University, 1732 Deogyeong-Daero, Giheung-Gu, Yongin-Si, Gyeonggi-do 17104, Korea; seunghyun@khu.ac.kr
[3] Financial Management Division, Kyung Hee University, 26 Kyungheedae-ro, Dongdaemun-gu, Seoul 02447, Korea; khlsh@khu.ac.kr
[4] School of Architectural Engineering, University of Ulsan, 93 Daehak-Ro, Ulsan 44610, Korea
* Correspondence: sky9852111@ulsan.ac.kr; Tel.: +82-52-259-2788

**Abstract:** In recent years, natural disasters and climate abnormalities have increased worldwide. The Fifth Assessment Report (2014) of the Intergovernmental Panel on Climate Change warned of extreme rainfall events, warming and acidification, global mean temperature rises, and average sea level rises. In many countries, changes in weather disaster patterns, such as typhoons and heavy rains, have already led to increased damage to buildings. However, the empirical quantification of typhoon risk and building damage due to climate change is insufficient. The purpose of this study was to quantify the risk of building loss from typhoon pattern change caused by climate change. To this end, the intensity and frequency of typhoons affecting Korea were analyzed to examine typhoon patterns. In addition, typhoon risk was quantified using the Korean typhoon vulnerability function utilized by insurers, reinsurers, and vendors, the major users of catastrophe modeling. Hence, through this study, it is possible to generate various risk management strategies, which can be used by governments when establishing climate change policies and help insurers to improve their business models through climate risk assessment based on reasonable quantitative typhoon damage scenarios.

**Keywords:** climate change; typhoon; catastrophe model; typhoon vulnerability function; risk analysis

## 1. Introduction

Climate change is expected to have serious consequences in a wide range of areas. It is expected to affect extreme weather events in the short term, as well as generating long-term effects such as disease spread and rising sea levels. Extreme weather events could include heat waves, cold waves, windstorms such as hurricanes, heavy rains, floods, a lack of precipitation, and drought. Many regions have suffered from the fatal effects of recent extreme weather events. Such extreme weather events have, of course, always been part of human history. However, recent extreme weather events have become greater in frequency and intensity than those in the past, and the potential for damage has increased rapidly.

Additionally, the current pattern of tropic cyclones is so different from past patterns that they are called super typhoons or super hurricanes. For instance, Typhoon Haiyan occurred in 2013 and became known as Super Typhoon Yolanda, as it was the most extreme tropical cyclone recorded on land. Its severe rain and winds made it difficult for South Asian nations to recover from the shattering damage of about USD 300 billion [1]. In the United States in 2017, three powerful hurricanes (Hurricanes Harvey, Maria, and Irma) caused tremendous damage. The total damage from these

hurricanes was about USD 293 billion, with Harvey causing USD 125 billion in damage, Maria causing USD 90 billion, and Irma USD 77.6 billion worth of damage [2]. In addition, Hurricane Katrina, which occurred in 2005, was one of the most damaging natural disasters in United States history. The heavy rain and strong winds generated by hurricanes have caused US Gulf Coast cities to suffer about USD 180 billion in direct and indirect damage [2]. In Europe, economic losses of around EUR 13 billion were incurred in 1999 due to the record rain and winds of the European storms Anatol, Lothar, and Martin [3].

However, despite these historic events and record damage, there are still debates about climate change and tropical cyclone patterns. Even though many studies have argued that climate change has affected tropical cyclones, other research argues that the evidence for this is poor. For instance, though some have asserted that the intensity of tropical cyclones gradually increases as the climate warms up [4–6], others argue that this increase is within the natural range of fluctuation in the frequency or severity of tropical cyclones in long-term climate observations [7]. Depending on the region, long-term climate observations may not be of sufficient duration to determine how climate change affects tropical cyclones, or the effects may not be clear. It is also difficult to predict how future activities will impact climate change. However, there is evidence that the damage caused by extreme weather events, especially tropical cyclones, is increasing every year [8]. Other studies have shown that this trend is damaging more people and assets, and the damage will be even greater given the high coastal population and property density of many cities, and reduced woodlands [9,10]. While these studies do not adequately rule out damage due to increased social vulnerability (e.g., income and population), it is difficult to view this as only an increase in extreme weather events and tropical cyclones. The trend is clear [11]. Therefore, we will analyze the intensity and frequency of typhoons that have affected Korea for a scientific and quantitative examination of the impact of climate change on typhoons. In addition, this study will assess the risk of building loss to quantify the damage caused by changes in typhoon patterns due to climate change.

## 2. Literature Review

### 2.1. Climate Change and Economic Impact

The United Nations Intergovernmental Panel on Climate Change (IPCC) warns against climate change in its 5th Assessment Report (AR5). Compared to pre-industrial levels, the report estimates that global average temperatures will rise by more than 1.5 °C in all scenarios by 2100. In addition, warming will continue as greenhouse gas emissions continue, and moreover, it is likely to exceed 2.0 °C in many scenarios. Additionally, the World Bank (2014) has a similar outlook. Global warming is inevitable due to greenhouse gases in the Earth's atmosphere, and the temperature will be 1.5 °C higher than before industrialization. Without reasonable steps to reduce greenhouse gas emissions, the planet is expected to warm up by up to 2 °C by the middle of the century and up to 4 °C by the end of the century [12]. Furthermore, Stern (2006) reported that in the absence of measures to reduce emissions, greenhouse gas concentrations would reach twice the pre-industrial levels in early 2035, raising the Earth's temperature by nearly 2 °C. This warming is expected to change the water cycle around the world, increasing the difference between wet and dry regions. As the heat expands into the deeper oceans, the ocean's circulation pattern will change and continue to warm, and the Earth's glaciers will decrease. Due to the reduced glaciers, the global average sea level is likely to rise more quickly than the rate of rise over the last 40 years (IPCC 2004). As mentioned above, many studies and research papers show that global climate change is certain and will increase, and warn against the side effects of warming [13].

The literature on the economic impact of climate change is as follows. The IPCC Fifth Assessment (2014) reports that increasing warming above 3 °C will result in a loss of 0.2% to 2.0% of annual GDP (gross domestic product), although estimates of damage vary widely from country to country [14]. The IPCC expects further acceleration of the occurrence of damage if warming exceeds 2 °C, but

these effects will be difficult to realize over the next 30 years. Moreover, if warming exceeds 2 °C, negative returns are expected from various portfolios [15]. Dietz and Stern (2014) estimate that when global warming reaches the 4 °C level, annual economic output will decrease by 50% compared to that without warming. They estimated a warming of around 3.5 °C by 2100 [16]. Stern (2006) estimated that over the next two centuries, global warming scenarios between 2.4 and 5.8 °C would result in a mean loss of about 5% (up to 20% in some regions) of global yearly GDP by 2100. These calculations indicate that no action has been taken on global warming, and the costs are expected to increase by more than 20% of GDP, given the wide range of risks and impacts. In addition, with simple extrapolation, it was estimated that extreme weather could damage 0.5% to 1% of global GDP by the middle of the century [13]. Mendelsohn et al. (2000) studied the potential damage using a global warming scenario (an increase of 2.5 °C by 2010), and estimated that the total market impact cost would not exceed 0.1% of GDP in 2100. Market impact may vary based on latitude. For example, in low latitude countries, warming increases damage. On the other hand, income is expected to increase at higher latitudes. However, if global warming is above 2.0 °C, it is expected that the benefits will decrease and the damage will increase. They also found that damage in a global warming scenario (2.0 °C increase by 2060) would be expected to have an aggregate impact of 0.3% damage to GDP in 2060. The study estimated that with warming of 2.0 °C by 2060, most of the damage would occur in agriculture, and the damage would vary widely from country to country [17]. As these studies show, climate change is predicted to have a significant impact on future economic growth and living standards. Losses may vary by region, but the damage is expected to increase globally. In addition, severe weather phenomena are expected to add to the damage.

### 2.2. Climate Change and Losses from Natural Disasters

The increase in damage caused by natural disasters is closely related to the growth of population and wealth. This is because the world's population is increasing every year, and wealth is also growing. The annual damage caused by natural disasters may be linked to these increases in wealth and population. Therefore, to objectively quantify climate change and the increase in damage from natural disasters, increases in wealth and population must also be considered [18]. To this end, many studies have examined climate change and damage after normalization for population and wealth changes. Nordhaus (2010) argued that since 1900, losses from hurricanes in the United States have increased significantly according to revised data only for GDP [19]. Changnon (2009) argued that insurance losses from hurricanes in the United States increased between 1952 and 2006 but that the growth was concentrated in the western United States and is believed to be due to recent increases in population and wealth in this region [20]. He also examined a study of insurance losses due to hail in the United States since 1992. The amount of insurance losses due to hail has increased, but this is attributed to increased exposure and vulnerability to hail due to the expansion of urban areas. There was no change in the frequency of major hail storms [9]. Chang et al. (2009) detailed a rise in flood loss through a flood loss survey (since 1971) in six cities in Korea. The cause of the increase in flood damage was found to be related to the increase in population, as well as summer rainfall and deforestation [10]. Schmid et al. (2009) found that there was a clear trend in US hurricane losses. However, this trend appeared after 1970 (it was not seen in the entire record dating back to 1950) and was found only after adjustments for wealth and population [21]. Fengqing et al. (2005) investigated flood damage in the Xinjiang Autonomous Region of China and determined that flood damage had increased since 1987. However, he pointed out that reserves and flood control structures, not heavy rains caused by climate change, were responsible for the increase in flood damage [22]. Changnon (2001) reported increased damage according to normalized data due to strong winds, rainfall, lightning, hail, and tornadoes since 1974 in the western United States. Nevertheless, the study also showed increased losses according to normalized data even in areas with reduced thunderstorm activity, suggesting that socioeconomic factors contributed to this trend [23]. Miller et al. (2008) analyzed the loss data for climate disasters around the world after revising them, taking into account wealth and population growth. Their main

findings were that since 1970, losses from climate disasters have increased, but this trend does not extend back to 1950. In addition, the authors believe that the increase in losses from climate disasters is due to the hurricane damage in the United States in 2004 and 2005 [24].

As with previous studies, wealth and population are important considerations in the study of the relationship between climate change and losses from natural disasters. This is because increases in population and wealth are important contributors to increased natural disaster damage. It is true that studies of natural disaster damage caused by climate change are difficult due to the close relationship between wealth, population, and natural disaster damage. Therefore, in this study, the vulnerability function was used to exclude the interference of wealth and population for the quantitative study of climate change and losses from natural disasters only. In addition, the existing studies judge the increase and decrease in damage amounts from natural disasters, and thus, it is difficult to quantitatively study building damage due to climate change. Therefore, this study divided buildings into three groups by occupancy (commercial, industrial, and residential) and considered the risk of building loss due to climate change for each category.

## 3. Framework of Study

The purpose of this study was to quantitatively prove climate change and typhoon changes. To achieve this, the study consisted of two parts. First, this study investigated typhoons that have affected Korea and analyzed the intensity and frequency of typhoons and changes in typhoon patterns.

Second, this study quantified the risk of building loss due to changes in typhoon patterns as a result of climate change. To quantify typhoon risk, this study used the Korean typhoon vulnerability function of major users of catastrophic (CAT) modeling: insurers, reinsurers, and suppliers. The buildings were divided into commercial, industrial, and residential types for analysis. As shown in Figure 1, this study was limited to typhoons that affected S. Korea from the 1970s to 2010s. Therefore, the research scope of this study was limited to S. Korea and reflected the architectural design standards and planning strategies of S. Korea. The results may be different in countries with different geographic or architectural design standards and planning strategies from S. Korea.

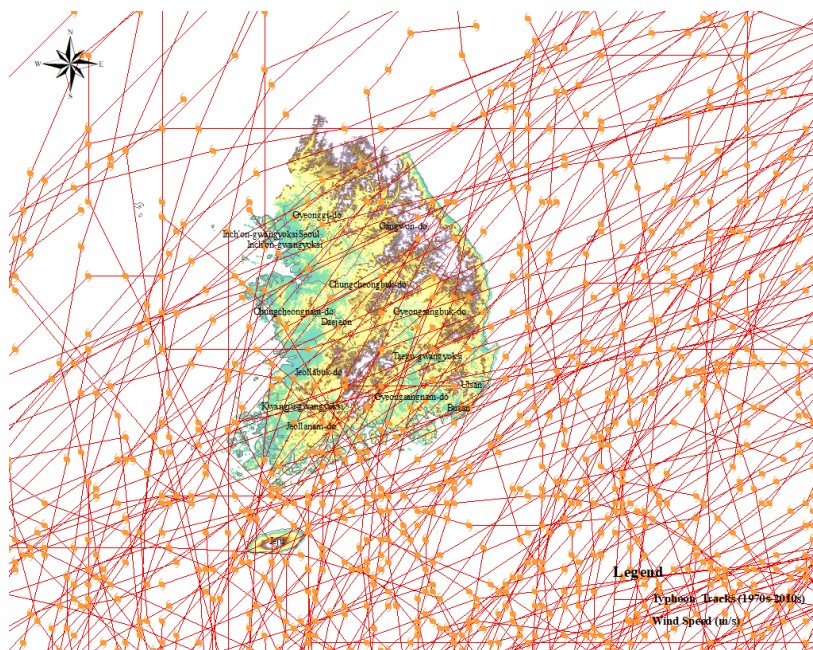

**Figure 1.** The typhoon that affected S. Korea (1970s to 2010s).

## 4. Typhoon Patterns

This part will discuss the frequency and severity of typhoons. Since the amount of risk is determined by the product of frequency and severity, both frequency and severity play an important role in risk determination. Therefore, frequency and severity were examined separately for detailed investigation.

Data on typhoons were obtained from the Korea Meteorological Administration (KMA, Seoul, Korea). The KMA was established in 1949 as a Korean government agency providing meteorological services and is responsible for monitoring the weather system and distributing and storing related information. This study collected data on the number of typhoons and the wind speeds that affected Korea between 1973 and 2019 from the KMA.

*Frequency and Severity of Typhoons*

The frequency of typhoons by year is shown in Figure 2. Korea was affected by an average of 3.3 typhoons annually, with a standard deviation of 1.5. The minimum number of typhoons generated during the survey was zero, and the largest number of typhoons affecting Korea was seven. The linear regression model of typhoon frequency is $y = 0.0036 \times \text{Year} - 3.8831$. The $R^2$ value is 0.0012. The slope of this model shows that the relationship between the year and number of typhoons is positive, indicating that the number of typhoons has increased slightly each year. However, the $R^2$ value shows that the relationship between the year and number of typhoons is very weak, since the value is extremely low. Hence, it would be difficult to conclude that climate change has a clear impact on the number of typhoons.

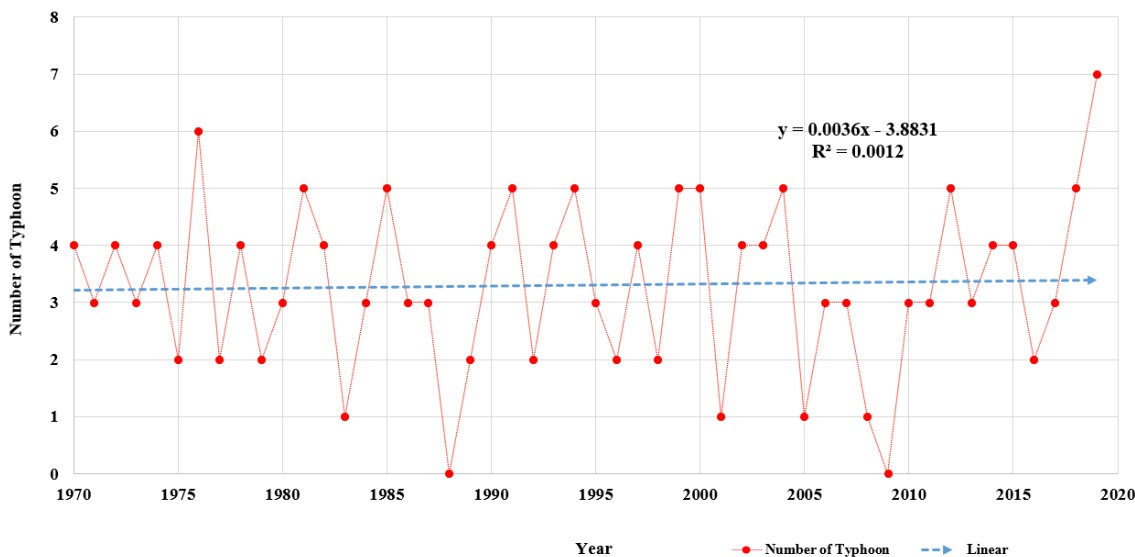

**Figure 2.** Number of typhoons by year.

The maximum wind speed of typhoons by year is shown in Figure 3. The wind speed was based on 10 min sustained wind speed. When a typhoon affected Korea, the highest wind speed recorded by 96 meteorological stations was considered as the maximum wind speed. After collecting the maximum wind speed by typhoon, the maximum wind speed was determined by year. The average maximum wind speed was 27.6 m/s, with a standard deviation of 8.4 m/s. The highest maximum wind speed was 51.1 m/s, and the lowest was 17.3 m/s.

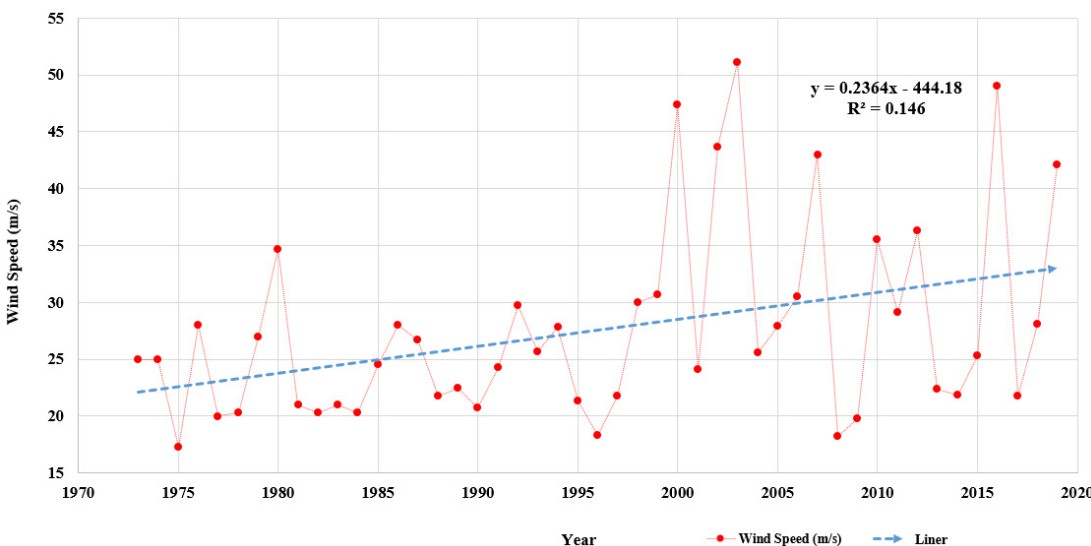

**Figure 3.** Maximum wind speed of typhoons by year.

The linear regression model of typhoon severity is y = 0.02364 × Year − 444.18. The $R^2$ value is 0.146. The slope of this model shows that the relationship between the year and maximum wind speed is positive, indicating that the maximum wind speed has increased each year. Furthermore, the $R^2$ value shows that the year and the maximum wind speed have a weak quantitative linear relationship.

Based on the typhoon data from the KMA, this study investigated the frequency and severity (max. wind speed) of typhoons by year. The frequency of typhoons was found to increase only slightly from year to year, but due to the low $R^2$ value, the explanatory power is low and not significant. However, the severity of typhoons appears to have increased year by year. Although the $R^2$ value is relatively small, it is a weak positive relationship and is sufficient to show a potential trend. This suggests that the frequency of typhoons does not increase every year, but the severity does. Thus, it is possible that the risk from typhoons has increased due to increasing severity.

However, the data of the KMA used in this study were recorded for about 50 years from 1973 to 2019. Although this period is a period for which the KMA's current national and regional data are available, it is considered to be a short period for concluding that the severity of typhoons is gradually increasing. Therefore, it is necessary to keep an eye on the trend through additional data collection.

## 5. Calculation of Increased Typhoon Risk

This section quantifies the risk of building loss due to changes in typhoon patterns resulting from climate change. This study adopted the Korean typhoon vulnerability function used by insurers, reinsurers, and vendors to quantify typhoon risk. The assessment of vulnerability to typhoons is a significant part of the typhoon risk assessment model. It is the vulnerability curve or vulnerability function that is used for this vulnerability assessment. The vulnerability function is expressed by quantifying the vulnerability of the building. The vulnerability function for typhoons explains the correlation between the average loss ratio, wind speed, and inventory information of various buildings, and determines the loss scale. The average damage ratio is the total amount of damage incurred by a building due to a typhoon divided by the total cost of the building. Therefore, the average damage ratio is used as a measure of a building's vulnerability to typhoons. For example, a high average damage ratio indicates that the damage is large due to a high vulnerability to typhoons. The vulnerability function was used because it quantifies the loss ratio for typhoons through various damage indicators such as the inventory information of buildings and wind speed, thus preventing the distortion of damage by wealth and population found in previous studies [25]. The catastrophic (CAT) model has been developed and used by a number of initiatives, global administrations, and private interests as a risk assessment tool for

scientifically assessing, responding to, or mitigating natural disaster risk. For example, public models include HAZUS Multi-Hazard in the United States, RiskScape in New Zealand, New Multi-Hazard and Multi-Risk Assessment Method (MATRIX) in Europe, and Central America Probabilistic Risk Assessment in South America. Vendor models include Risk Management Solutions, Applied Insurance Research, and Risk Quantification and Engineering, which develop and use models for natural disasters and other risks as business models. Primary and reinsurance companies quantify risks from natural disasters by actively using in-house or vendor models. They use the CAT model to manage their portfolios, capital, business preferences, and holding strategies and for capacity monitoring based on the quantified risks of natural disasters [25,26]. The CAT model generally consists of four parts (i.e., a hazard module, exposure module, vulnerability module, and financial module). Each module has an independent function, and the operation of the modules proceeds sequentially. First, the hazard module generates events and calculates local intensity to physically define the events and describe the severity and frequency of natural disasters. Second, the exposure module embodies inventory and geographic information for a building. Third, the vulnerability module provides the loss ratio based on the vulnerability function, which determines the average loss ratio based on wind speed and building inventory information. Lastly, the financial module applies certain insurance factors, such as deductibles and liability limits, to calculate financial losses [25]. For instance, in the hazard module, the severity and frequency of typhoons are defined through simulation according to the characteristics of past typhoons in a specific area. In the exposure module, wind speed is determined according to the inventory characteristics and geographic characteristics of buildings in that specific area. Depending on the determined wind speed, the vulnerability module computes the loss amount through the vulnerability function. The calculated loss amount is calculated by considering insurance conditions in the financial module.

Figure 4 illustrates the vulnerability functions for each model. This study used the vulnerability function of loss ratio for wind speed and occupancy. Occupancy, a representative relative vulnerability factor, was used to reflect the vulnerability of building inventory. Occupancy is used in risk management and risk assessment models among other building inventory information. Occupancy also refers to a similar accounting policy in insurance that categorizes buildings as industrial, residential, and commercial. This classification of buildings according to occupancy refers to building units with similar physical and financial characteristics. This study also adopted the occupancy classification and divided buildings into industrial, residential, and commercial groups.

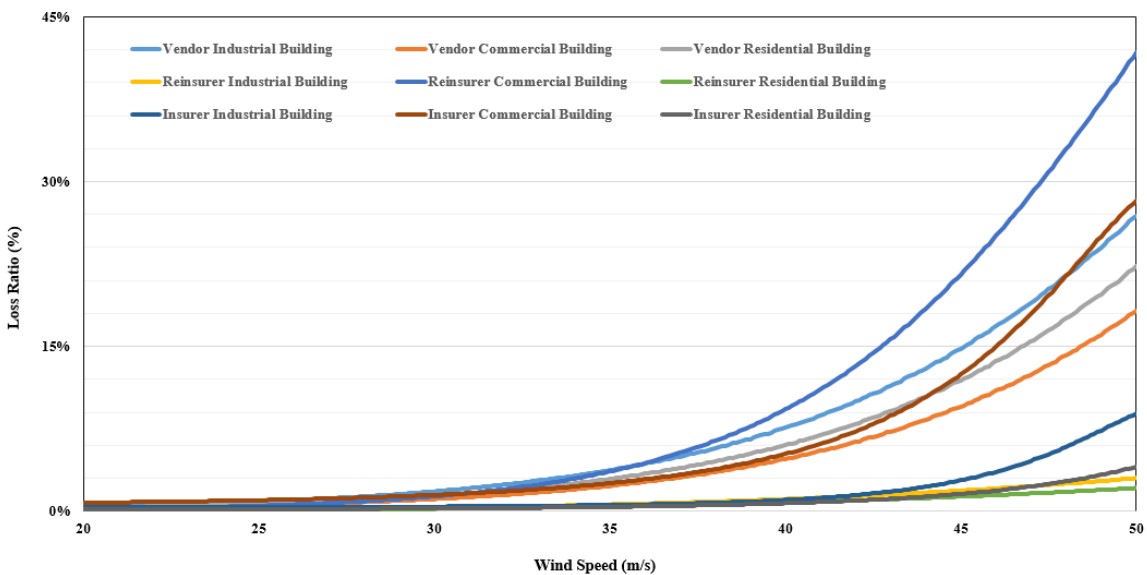

**Figure 4.** Vulnerability functions for each model.

*Results of Analysis*

The analysis results are shown in Tables 1–3. To clearly show the increase rate, the five decades from the 1970s, when typhoon data began to be recorded, until the recent 2010s were compared. Additionally, as seen in Equation (1), each decade showed a numerical increase or decrease compared with the 1970s. The period in Equation (1) refers to the 1980s, 1990s, 2000s, or 2010s.

$$\text{The rate of increase each decade (\%)} = (\text{period} - 1970\text{s})/1970\text{s} \tag{1}$$

**Table 1.** Result (average) summary of each model for industrial buildings.

| Period | Vendor | | Reinsurer | | Insurer | | Average | | Increase Rate | CV |
|--------|--------|--------|--------|--------|--------|--------|--------|--------|--------|--------|
| | Ave. | ST. | Ave. | STD. | Ave. | STD. | Ave. | STD. | | |
| 1970s | 0.78% | 0.35% | 0.12% | 0.07% | 0.35% | 0.02% | 0.42% | 0.15% | - | 0.4 |
| 1980s | 0.97% | 0.92% | 0.15% | 0.15% | 0.36% | 0.06% | 0.50% | 0.38% | +19% | 0.8 |
| 1990s | 1.04% | 0.58% | 0.16% | 0.11% | 0.37% | 0.03% | 0.52% | 0.24% | +26% | 0.5 |
| 2000s | 7.66% | 9.11% | 0.94% | 1.05% | 1.99% | 2.70% | 3.53% | 4.29% | +750% | 1.2 |
| 2010s | 4.80% | 6.98% | 0.63% | 0.81% | 1.22% | 2.07% | 2.22% | 3.29% | +434% | 1.5 |

**Table 2.** Result (average) summary of each model for commercial buildings.

| Period | Vendor | | Reinsurer | | Insurer | | Average | | Increase Rate | CV |
|--------|--------|--------|--------|--------|--------|--------|--------|--------|--------|--------|
| | Ave. | STD. | Ave. | STD. | Ave. | STD. | Ave. | STD. | | |
| 1970s | 0.61% | 0.28% | 0.08% | 0.05% | 0.17% | 0.03% | 0.29% | 0.12% | - | 0.4 |
| 1980s | 0.76% | 0.72% | 0.10% | 0.11% | 0.18% | 0.07% | 0.35% | 0.30% | +22% | 0.9 |
| 1990s | 0.81% | 0.46% | 0.12% | 0.07% | 0.19% | 0.04% | 0.37% | 0.19% | +30% | 0.5 |
| 2000s | 6.22% | 7.50% | 0.66% | 0.74% | 0.99% | 1.24% | 2.63% | 3.16% | +819% | 1.2 |
| 2010s | 3.87% | 5.73% | 0.45% | 0.57% | 0.62% | 0.92% | 1.65% | 2.41% | +476% | 1.5 |

**Table 3.** Result (average) summary of each model for residential buildings.

| Period | Vendor | | Reinsurer | | Insurer | | Average | | Increase Rate | CV |
|--------|--------|--------|--------|--------|--------|--------|--------|--------|--------|--------|
| | Ave. | STD. | Ave. | STD. | Ave. | STD. | Ave. | STD. | | |
| 1970s | 0.47% | 0.22% | 0.45% | 0.27% | 0.95% | 0.18% | 0.63% | 0.22% | - | 0.4 |
| 1980s | 0.60% | 0.56% | 0.68% | 0.93% | 1.06% | 0.49% | 0.78% | 0.66% | +24% | 0.9 |
| 1990s | 0.64% | 0.36% | 0.69% | 0.50% | 1.09% | 0.30% | 0.80% | 0.38% | +28% | 0.5 |
| 2000s | 5.02% | 6.11% | 10.91% | 14.27% | 7.23% | 9.09% | 7.72% | 9.82% | +1133% | 1.3 |
| 2010s | 3.10% | 4.67% | 6.27% | 11.03% | 4.43% | 7.13% | 4.60% | 7.61% | +634% | 1.7 |

The differences in the average loss rates for each model ranged from 0.38% to 6.72%. The average loss ratio for each model is as follows. In the case of the Vendor model, industrial buildings have 3.05%, commercial buildings have 1.96%, and residential buildings have 2.45%. In the case of the Reinsurer model, they are 0.40% for industrial buildings, 3.80% for commercial buildings, and 0.28% for residential buildings. In the case of the Insurer model, they are 0.86% for industrial buildings, 2.95% for commercial buildings, and 0.43% for residential buildings.

The vulnerability function is generally developed based on the available or actual losses of the model developers. However, losses may differ due to the differing capital, business preferences, and portfolios of each model developer. These factors will make a difference, even if Korea's vulnerability function is the same. To compensate, the insurance industry typically compares and validates the results of two or more models. Therefore, this study improved the reliability of the results by comparing multiple models from three different fields. This study averaged the results of three models and compared the increase and decrease by decade. Table 1 describes the result (average) summary of each model for industrial buildings. The increase rate has grown gradually over the decades compared with the 1970s, increasing 307% on average and 434% in the recent 2010s. In the 2000s, it ballooned to 750%. This was due to Typhoons Maemi (2001) and Rusa (2002), which caused the most damage in history, with the highest wind speeds in Korea. The coefficient of variation (CV) is used to compare data with

different units of measurement. The CV is the standard deviation divided by the arithmetic mean. The larger the CV, the larger the relative difference. The CV also steadily increased every 10 years, indicating that the difference in the intensity of the typhoons (maximum wind speed) increased. Compared to the past, typhoons of various intensity are occurring now, indicating that the current typhoons can exhibit more diverse damage categories than the past typhoons. These values (i.e., average, increase rate, and CV) prove that the loss incurred by buildings is greater than that in the 1970s, and this proves that typhoons are more serious than in the past. Table 2 represents the result (average) summary of each model for commercial buildings. The increase rate has grown progressively over the decades since the 1970s, reaching 455% on average and 634% in the recent 2010s. During the 2000s, it intensified to 1133% due to Typhoons Maemi (2001) and Rusa (2002). The CV rose steadily every period and was projected to vary as the maximum wind speed escalated each decade. This shows that current typhoons produce more varied intensity than past typhoons. Table 3 shows the result (average) summary of each model for residential buildings. The increase rate has grown over the decades since the 1970s, reaching 337% on average and 476% in the recent 2010s. During the 2000s, it increased dramatically to 819% owing to Typhoons Maemi (2001) and Rusa (2002). The CV rose increasingly each decade and was predicted to vary as the maximum wind speed rose. These demonstrate that current typhoons are more severe than past typhoons.

## 6. Discussion

This study was a quantitative study of the changes in tropical cyclones caused by climate change. It investigated the typhoons affecting Korea in the past and analyzed their intensity and frequency, as well as changes in typhoon patterns. The analysis results showed that the frequency of typhoons did not increase, but their severity did increase. This indicates that the risk from typhoons is growing due to their increasing severity.

To quantify the increased typhoon risk, this study used the vulnerability function of the CAT model, which is not affected by wealth and population. As a result, it was shown that the risk from typhoons increased gradually during the 1970s, 1980s, 1990s, 2000s, and 2010s. On average, the risk to industrial buildings increased by 307%, the risk to commercial buildings by 455%, and the risk to industrial buildings by 337%. The increase rate for commercial buildings was the largest, which is attributed to the fact that commercial buildings consist of more diverse buildings than other building types. In addition, the 2000s displayed the largest increase rate due to the influence of Typhoons Maemi and Rusa. These two typhoons were the largest typhoons in Korea, but they were included in the analysis because they are generally considered to be 15–30 return period typhoons. The analysis results show that the risk from typhoons has increased significantly each year.

For this reason, new strategies are required to respond to changing circumstances and increasing risks. The insurance industry is critically sensitive to hurricanes. For example, 11 insurers in the US were bankrupted by Hurricane Andrew (1992). Therefore, a review of pricing, policy conditions, and reinsurance for increased loss risk is essential. In terms of pricing, there is a need to raise current premiums and modify the current probable maximum loss and limit of liability. Moreover, changes in acquisition, retention, and accumulation management strategies are inevitable due to the changed pricing. Policy conditions require a review of the scope of coverage of existing insurance policies. In facultative and treaty reinsurance arrangements, new strategies for excess loss and layering for enlarged risk are needed. Furthermore, it is crucial to calculate premiums through accurate quantification of the weighted risks, which can be performed through appropriate CAT models. Increased risk, on the other hand, may also be a new opportunity because it requires active risk transfer from the government or private sector, leading to a boom in insurance coverage. The introduction of CAT bonds is also desirable to hedge losses from catastrophic events in developing countries such as Korea. CAT bonds are used in developed countries such as the US and Germany to distribute reinsurance functions through the generation of bonds when the risk for insurance companies exceeds their acquisition capacity.

Governments also need to strengthen architectural design standards and codes to create a sustainable building environment that can withstand extreme weather disasters. Additionally, a separate management guide is required for older buildings constructed with past building codes. Maintaining infrastructure, lifelines, and transportation systems during hurricanes is critical to reducing human and property damage, requiring a more advanced management system. Storm and flood hazard areas should also be mandated to actively transfer the risk of loss through mandatory subscription to storm and flood insurance. This should enable local communities and residents to respond appropriately to changing circumstances.

## 7. Conclusions

Many historical event and damage analysis studies continue to debate climate change and its effect on tropical cyclones. Therefore, this study aimed to analyze the intensity and frequency of typhoons by examining typhoons affecting Korea for scientific and quantitative research on climate change and typhoon changes and to quantify the damage caused by these changes. In addition, the risk of building loss was quantified. The results show that the severity of typhoons increases year by year, and thus the risk from typhoons also increases year by year. This suggests that climate change is affecting typhoons. Hence, it is necessary to consider government and industry responses to climate change and risk reduction.

On the other hand, since this study only considers wind speed and occupancy, the results may vary when additional inventory information is used. Further research using inventory information from multiple buildings is needed. Because different results can be obtained for different geographic regions of the Korean Peninsula, further research is needed to analyze the results. In particular, the results will differ in the southern part of the main typhoon route. Moreover, this study may not represent other countries, as the research area is limited to Korea. Countries with more coastal development than Korea may be more vulnerable to typhoons, and countries with greater building capacity or maintenance management systems may be better able to adapt to storms. Adaptation policies for climate change and risk reduction have not been modeled. Damage can be reduced through active climate change adaptation policies and programs by individuals or institutions, and comprehensive research that includes this is needed.

**Author Contributions:** Conceptualization, J.-M.K.; Data curation, J.-M.K.; Funding acquisition, K.S.; Investigation, S.S., S.L.; Methodology, J.-M.K.; Software, S.S., S.L.; Validation, J.-M.K., K.S.; Writing original draft, J.-M.K.; Writing review and editing, J.-M.K., K.S., S.S., and S.L. All authors have read and agreed to the published version of the manuscript.

**Funding:** This work was supported by the 2020 Research Fund of University of Ulsan.

**Conflicts of Interest:** The authors declare no conflict of interest.

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
