# Peer review of "Cost of Climate Change: Risk of Building Loss from Typhoon in South Korea"

_sustainability, doi:10.3390/su12177107_

Round 1

Reviewer 1 Report

The submitted manuscript aims at quantitatively estimate the risk of building loss from typhoon pattern change caused by climate change. Although the manuscript faces with a crucial topic and presents a very interesting collection of data about the number of typhoons by year and the maximum wind speed by year, the period of observation is too short to claim that there is a constantly increasing risk of building loss due to climate change and the scientific contents of the manuscript is poor. By observing the data, it is evident that there is a sudden change from the 2000s: hence, in the reviewer’s opinion, one may just claim that in the first two decades of the XXI century the severity of the typhoons has been higher than during the preceding 30 years. Not all the information given by the authors are coherent the conclusions of the submitted manuscript and some points are not described in a sufficiently detailed way. Furthermore, some sections of the manuscript are hardly readable. Hence, English correction of the revised version by a native speaker is recommended.

  • Section 2.1 can be written in a slightly way. It is slightly confusing to read it as is. Please, revise this section.

  • Section 4.1: The linear term of the best-fit line obtained by the linear regression of the numbers of typhoon by year is almost negligible. In my opinion, it is not possible to claim that there is a clear effect on the number of typhoon of climate change. A comment should be added.

  • Section 4.1: It is evident that the linear fit with the experimental data is poor. The linear regressions might be of interest because they give information about a potential trend, but the values reached by R2 do not give any contribution to improve the quality of the manuscript.

  • Pag 5; lines 192-193: “The maximum wind …. wind speed”. Please, rephrase this sentence to make it clearer.

  • The title of Section 5 might lead to a misleading interpretation. Please, change the title.

  • Page 6; lines 216-244. This part is confusing. I think that is should be re-written by better explaining what is a vulnerability function and by better describing the main features of the CAT model.

  • It is not clear what the terms in equation 1 represent.

  • Page 7; lines 255-256: Where are reported the average loss ratios for each model?

  • Page 7: lines 263-265. By observing Figure 2 and Tables 1, 2, and 3 it is evident that the increase rate has not grown gradually over the decades, whereas it had a sudden growth from the 200s.

  • Page 7: lines 269-271. The Coefficient of Variation is a non-dimensional measure of the dispersion. It is not clear why the authors claim “This shows that there is currently a greater severity of typhoons than in the past”. Furthermore, it is not true that the CV “grew steadily each decade”, as it is evident from Tables 1, 2, and 3.

Reviewer 2 Report

The authors address a very interesting topic that is progressively becoming more urgent in coastal areas all over the world. The text is well written, documented and referenced. The markup presents a moderate amount of proof-readers comments. Specific comments are provided below.

  1. The fragment on the current pattern of tropic cyclones starting in line 40 could (and should) be a separate paragraph.
  2. References for damages quoted in lines 44 and 47 are missing.
  3. The last paragraph of the Introduction would benefit from some reflection and/or references regarding density of the coastlal areas (since risk is the intersection of assets, threats, and vulnerabilities) and the research focuses on built environment. It will allow to eliminate stating the obvious in lines 132 and 134-135.
  4. The previsions by Stern quoted in lines 101-102 were actually more alarming and some of the more recent provisions are even more pesymistic; 5% loss og global yearly GDP by 2100 is not exactly 5% (up to 20% in some regions) loss of global GDP every year for the next 100 years; consider revising & updating. 
  5. Line 112: I think (and hope) you meant 2.0 degrees, not 20 degrees.
  6. Line 121: I'm not sure whether it's sensible to give all the credits for increasing wealth to industrialisation; consider revising.
  7. Apart from architectural design standards, it would be useful to refer to planning strtategies either general or specific for the region.

Given the interdisciplinary character of the journal, the paper would benefit from an illustration showing the range of the typhoons: the areas subjected to the study and their densities. 

Hopefully you will find the above comments useful.

Reviewer 3 Report

The paper deals with about the risk of building loss from typhoon in South Korea.
The information found for carrying out a depth descrition have been very poor but in any way the reviewer is aware in the difficulty of collecting the data. However, the manuscript is well written and organized. The conclusions are in line with the results. It is suggested for a possible pubblication.
The activity could be thorough, in a next work, describing better the structural damage in the buildings.

minor issue

Improve the quality of all figures.

Round 2

Reviewer 1 Report

The overall quality of the paper and, in particular, its readability are strongly improved. However, although the treated topic is of primary interest and the presented data are extremely interesting, the scientific contents is poor in my opinion. I still think that, since the observation period is relatively short, one may claim that in the first two decades of the XXI century the severity of the typhoons has been higher than during the preceding 30 years, but not that there is a gradual increasing of the observed parameters. Furthermore, it is not yet clear how the values for the coefficient of variation recorded in Table 1, 2, and 3 are correlated with the typhoon intensity. 

Round 3

Reviewer 1 Report

The authors modified the manuscript trying to take into account all my observations. I believe that the overall quality of the article is sufficient to be published.